# Beyond traditional stimuli: Validating AI-generated images for eliciting negative emotions in affect research

Hey Tou Chiu[1,2☯], Hoi In Sou[1,2☯], Yuen Wing Lam[1,2], Clayton Siu Fung Ng[2], Savio W.H. Wong[1,2]*

**1** Department of Educational Psychology, The Chinese University of Hong Kong, Hong Kong,
**2** Laboratory for Brain and Education, The Chinese University of Hong Kong, Hong Kong

☯ These authors contributed equally to this work.
* savio.wong@gmail.com

## Abstract

Studies of emotion often rely on standardized stimulus sets to elicit affective responses. Although established databases provide images with normative valence and arousal ratings, selecting suitable stimuli can be difficult when experiments require specific thematic or content constraints. This challenge is especially pronounced for negative stimuli, which are central to research on maladaptive emotions and behaviors in clinical contexts but are often scarce in necessary quantity or specificity. The present study evaluated the feasibility of using generative AI, specifically text-to-image generators, to create tailored negative and neutral affective stimuli. To assess whether these images can serve as alternatives to traditional stimuli, we compared their affective properties to those reported in standardized image databases. Across two studies, participants rated the valence and arousal of 160 and 200 AI-generated images. Our findings revealed that AI-generated negative and neutral images reproduced the characteristic inverse association between valence and arousal observed in standardized databases, with moderate to strong correlations between these dimensions. These results highlight the potential of generative AI as a practical methodological tool for creating customized affective stimuli aligned with specific research objectives and experimental designs.

## Introduction

A central aspect in the study of emotions involves examining individuals' responses to controlled stimuli designed to elicit specific emotional responses [1]. The development and utilization of emotionally salient materials have become essential for accurately measuring behavioral responses (e.g., reaction time, accuracy) and physiological reactions associated with specific emotions [2]. Researchers employ various modalities of stimuli, including visual, lexical and auditory, to induce emotional

**Data availability statement:** The rating data of Study 1 and Study 2 are included as supporting information in table S2 and S3 respectively.

**Funding:** General Research Fund from the Research Grant Council Hong Kong (RGC Ref No. 14619919) Social Innovation and Entrepreneurship Development Fund (SIE Fund) KPF20QEP12.

**Competing interests:** The authors have declared that no competing interests exist.

responses [3,4]. Among these, visual stimuli (images) are the most widely used in behavioral and neuroimaging research because they require minimal linguistic knowledge and semantic processing, making them intuitive and particularly suitable for cognitive research of affective processing compared to textual or auditory stimuli [5–7]. However, before visual stimuli can be used as reliable emotional elicitors, researchers must carefully control stimulus content and physical properties (e.g., size, brightness and colour) tailored to specific research questions. Consequently, identifying appropriate visual stimuli aligned with particular experimental requirements remains challenging, as consistently noted across previous studies [2,8,9].

Visual stimuli in affective research mostly come from existing standardized image databases such as the International Affective Picture System (IAPS) [10], the Nencki Affective Picture System (NAPS) [9], and the Open Affective Standardized Image Set (OASIS) [11]. These databases offer images with a wide range of themes and content, along with normative ratings on affective dimensions. Valence, measuring the positive or negative nature of an affective experience, contrasts states of pleasure with displeasure. Arousal focuses on the distinct level of excitement induced [12]. The self-reported ratings on the affective dimensions are usually captured using the Self-Assessment Manikin (SAM) [13], whereby valence and arousal are represented by pictorial figures that are inserted along a 9-point scale. The SAM is a culturally and linguistically universal measuring instrument which targets affective responses associated with the stimuli ("How do you feel while viewing the picture") instead of the semantic knowledge ("Are the object or situations depicted good or bad?"). This technique has facilitated replicability through validation of affective images across languages and cultures (e.g., [1,5,14,15]) and is currently widely adopted as a standardized procedure for collecting normative ratings in affective databases. Importantly, the stimuli in these databases have been extensively used in various experimental paradigms in both behavioral and neuroimaging research (e.g., [16–19]).

Several limitations of existing standardized stimuli databases have been identified. First, the availability of stimuli within specific categories is often constrained [1,9]. For research requiring a high frequency of stimuli aligned with particular themes or content categories, broad-topic databases, such as IAPS, may fall short in providing sufficient suitable options for task-specific purposes [2]. This is particularly the case in studies of negative affect processing, an area that has historically dominated affective research because of the greater motivational and clinical relevance of negative emotions to behavior [20–22]. Suitable negative images are often difficult to obtain in large quantities and may require researchers to select specific stimuli and combine them as a set from multiple databases [23,24]. Additionally, as stimuli from these databases are predominantly natural photographs, inconsistencies in image quality and perceptual characteristics (e.g., color, size, brightness) can complicate the process of maintaining experimental control over visual stimuli. Furthermore, certain images in databases like the IAPS may feel outdated or contextually irrelevant, as the database was originally developed in a pre-internet era [25]. Thus, when research designs require specific valence, image content, precise control over perceptual attributes or image styles, and when these images are needed in a substantial quantity, it

becomes imperative to explore novel methodologies for generating stimuli tailored to affective research. Such approaches not only address the inherent limitations of existing database but also reduce the time-intensive process of searching for suitable stimuli.

### Current uses of artificial intelligence in stimuli development

Recent technological advancements have spurred the rapid growth of artificial intelligence (AI). Particularly, generative AI enables automatic creation of diverse content, encompassing texts, images, and videos, in response to user-provided prompts [26,27]. These innovations have encouraged researchers from diverse disciplines to leverage AI for generating materials tailored to their specific research paradigms. To date, generative AI has been employed to create visual and linguistic stimuli across various disciplines, ranging from the arts, linguistic to psychological research [28–30]. For example, Alzahrani et al. [30] examined the feasibility of using AI to generate auditory and written sentence stimuli and evaluated its acceptability and validity across three psycholinguistic experimental designs. Using Lovo AI, a text-to-speech tool and ChatGPT-3 for generating AI-generated sentences, Alzahrani et al. [30] showed that the quality of AI-generated psycholinguistic stimuli in English was perceived as comparable to or superior to those developed by experienced researchers. Although these stimuli were unable to consistently replicate established psycholinguistic effects, the study provided evidence of high acceptability, indicating that stimuli were perceived as human-like. For other types of stimuli, such as AI-generated faces, studies have shown that they are incredibly hard to distinguish from real-life faces (e.g., [31,32]). AI-generation techniques have been applied to introducing subtle changes in facial expressions to examine its impact on participants aesthetic ratings [33]. More recently, Tassinari [34] used Dall-E 2 [35] to generate specific stimuli tailored to study weight biases by generating average and overweight versions of facial stimuli, intended for use in the Implicit Association Test (IAT). These studies demonstrated that AI shows promise in developing stimuli comparable to those produced by humans and is increasingly adopted as a tool to modify or create specific stimuli to study psychological processes, making it a potentially valuable tool for use in experimental research.

Recent studies have further investigated the potential of generative AI to generate emotionally charged materials [28,29,36]. For instance, Demmer et al. [28] created visually abstract artworks using a random noise generator (RNG) and compared them to artworks created by human artists. Participants were asked to report the extent to which they experienced emotions while viewing both types of artwork. The results revealed that participants reported feeling emotions and ascribing intentions to the artworks, regardless of whether they were created by AI or human artists. This suggests that AI-generated artworks are capable of eliciting emotional responses in viewers. In another study, Azuaje et al. [36] developed a therapeutic writing tool incorporating text-to-image AI to generate artwork intended to positively distract users from negative emotions. The results indicated that while the tool contributed to improvements in some emotional outcomes, such as reductions in anger and sadness, it was less effective in addressing other emotions, such as anxiety or stress. Moreover, the intended positive distraction of the AI-generated images was inconsistent; some participants found the images negative and unsettling [36]. Although the inclusion of AI-generated artwork in the writing tool did not consistently help participants downregulate their negative emotions, the study demonstrated that AI-generated images can effectively evoke both positive and negative emotional responses in viewers.

### Research gap

Despite the increasing use of AI in stimuli generation, it is unclear whether generative AI is suitable for creating standardized emotionally provoking stimuli specifically tailored for experimental designs in affective research. While it seems that AI-generated artworks can readily evoke emotional responses in participants [28,29], no study has yet to systematically examine the affective dimensional properties of AI-generated images, particularly in naturalistic scenes. To establish that AI-generated emotional images can be used as a valid tool for emotion research, it is essential to investigate whether

visual stimuli created through generative AI can reproduce the normative valence and arousal patterns observed in standardized affective stimuli [10]. Standardized affective stimuli typically exhibit a "boomerang" or "U-shaped" distribution, where positive and negative stimuli are rated higher on arousal, while neutral stimuli tend to be rated lower [5,37,38]. If AI-generated affective stimuli demonstrate similar properties in these dimensions, AI could emerge as a viable tool for affective researchers, particularly for sourcing additional or highly specific stimuli. With the growing demand for large quantities of themed visual stimuli, and for stimuli tailored to specific experimental designs, exploring the potential of generative AI to complement existing standardized image databases is critical. Moreover, findings from this exploration will provide insights into the broader applicability and limitations of AI-generated stimuli within future affective research.

### Present study

The present study investigates the potential of generative AI in creating static negative and neutral visual stimuli for affective research. To our knowledge, this is the first study to utilize text-to-image generative AI to develop naturalistic scene stimuli tailored to specific experimental designs. Beyond generation, we sought to establish normative ratings using standardized validation procedures. The primary novelty of this study lies in its demonstration that AI tools can produce tailored emotional scenes that yield replicable and consistent affective ratings across independent samples. While previous studies have examined emotional response to AI-generated artwork [28], the inter-associations of affective dimensions (valence and arousal) in AI-generated naturalistic scenes remain unexplored. The scope of this study was intentionally limited to the negative-to-neutral valence spectrum for practical and clinical validity. Methodologically, incorporating all three image categories (positive, neutral, and negative) for within-subject ratings would significantly increase the number of trials. This could lead to participant fatigue and habituation effects. Furthermore, given that negative affect is central to understanding psychopathology, such as PTSD [39], depression [40,41], anxiety [42], prioritizing this spectrum allows for a more focused contribution to the dominant literature on maladaptive emotional processing [20–22]. Therefore, we prioritized data quality and clinical relevance over a full-spectrum valence investigation.

We conducted two image rating studies using AI-generated stimuli specifically developed for an executive function task [43]. In Study 1, we collected valence and arousal ratings from participants who had just completed a behavioral task using these images. This design aligns with standard practices where post-task ratings serve as a manipulation check to verify that stimuli eliciting the intended affect within specific experimental context (e.g., [44–46]). However, because prior exposure can introduce habituation effects, we conduct Study 2 with two independent groups of exposure-naïve participants. This second study provides a cleaner, normative set of ratings unconfounded by task demands. By reporting findings from both studies, we offer a comprehensive validation: Study 1 demonstrates the effectiveness of the stimulifollowing task engagement, while Study 2 establishes a generalizable benchmark for future research. All stimuli and rating datasets are available for research use upon request.

## Study 1

### Method

**Development of AI-generated stimuli.** A set of 160 images (80 negative and 80 neutral images) was developed for an executive function task as part of a larger study [43]. Each image was designed to feature a combination of two of four specific content categories: animal [A], people [P], tree [T] or vehicle [V], yielding six possible combinations: people-tree (PT), people-animal (PA), people-vehicle (PV), tree-animal (TA), tree-vehicle (TV) or vehicle-animal (VA). The distribution of images across these combinations is detailed in Table 1.

We used three text-to-image generation models: 1) Stable Diffusion, 2) Adobe Firefly and 3) Leonardo.Ai to generate these images. Stable Diffusion was released in 2022, which uses latent diffusion as a deep learning technique to generate images based on text inputs [47]. Stable Diffusion can be implemented through front-end platforms, like DreamStudio, that allows for additional processing functions that enable users to mask a specific image area and filling it with further

**Table 1. Number of images rated per category in Study 1.**

| Category | Number of Images | |
|---|---|---|
| | Negative | Neutral |
| PT | 20 | 10 |
| PA | 10 | 20 |
| PV | 10 | 10 |
| TA | 10 | 10 |
| TV | 10 | 20 |
| VA | 20 | 10 |
| Total | 80 | 80 |

Each image contained two content categories as indicated by: PT=people-tree, PA=people-animal, PV=people-vehicle, TA=tree-animal, TV=tree-vehicle, VA=vehicle-animal.

text prompts (i.e., "inpainting"). It allows users to extend an image beyond its original dimensions, again through prompts inputted to Stable Diffusion, to generate new content (i.e., "outpainting"). In this study, we used Stable Diffusion v1.6 within DreamStudio.

Adobe Firefly operates on a generative AI model and is trained on licensed content, including Adobe Stock and public domain images where copyright has expired [48]. Users can access Firefly through the web browser with an Adobe Creative Cloud account, and can utilize a range of features (e.g., text-to-image generation, generative recoloring and generative fill). This study used the Firefly Image 2 Model. Leonardo.Ai is an advanced AI image generator that can create impressive graphics in a short amount of time. It offers a range of fine-tuned models, two of which are Leonardo Diffusion XL and Leonardo Kino XL. These models were built on top of existing sophisticated models, to improve the quality of generated images, also with the purpose of tailoring the models to produce specific styles. Both models used Stable Diffusion XL 1.0 as their base model [49] and can be accessed through creating a free account on the web platform. Among the three AI models, the majority of our negative images were generated using Stable Diffusion and Leonardo.Ai, while our neutral images were mostly generated using Adobe Firefly.

Images were generated using text prompts that specified negative and neutral scenes, each incorporating two specific content categories. Both inclusive and exclusive prompts were used: inclusive prompts specified two of the four categories (A, P, T, V) while exclusive prompts omitted the remaining two categories to ensure that the image content aligned with intended criteria. In addition to excluding specific categories, prompts were refined to control various aspects of the image output, such as tone (e.g., "with grey skies or drizzles" to create a cooler tone and a more negative mood), and the background (e.g., "no tall buildings behind" to minimize background distractions). The resulting scenes varied based on the category combinations. For example, negative images depicted scenarios such as a *car* hitting a *person* on the road, causing injury or blood, or, a fallen *tree* trapping an *animal,* resulting in death or distorted figures. Negative prompts were generally centered around themes of accidents, injuries, violence, disasters and catastrophes, which are similar to those found in standardized databases like IAPS and NAPS. Neutral images, on the other hand, typically depicted a person or an animal in a natural setting with a tree or a car in the background, such as a *man* sleeping under a large *tree*. Fig 1 provides examples of text prompts for both a negative and a neutral image, illustrating how these prompts were used to generating scenes corresponding to specific content category combinations. Additional examples of text prompts and corresponding generation parameters are provided in S1 Table.

Generating suitable images often involved iterative adjustments and repeated refinements of the text prompts, as initial prompts did not always result in images that fully matched our expectations. Distortions were particularly common in scenes depicting human and animal faces or limbs. Additionally, the backgrounds of some scenes occasionally appeared overly stylized or unrealistic. To address these issues during the process of image generation, we utilized features within

 

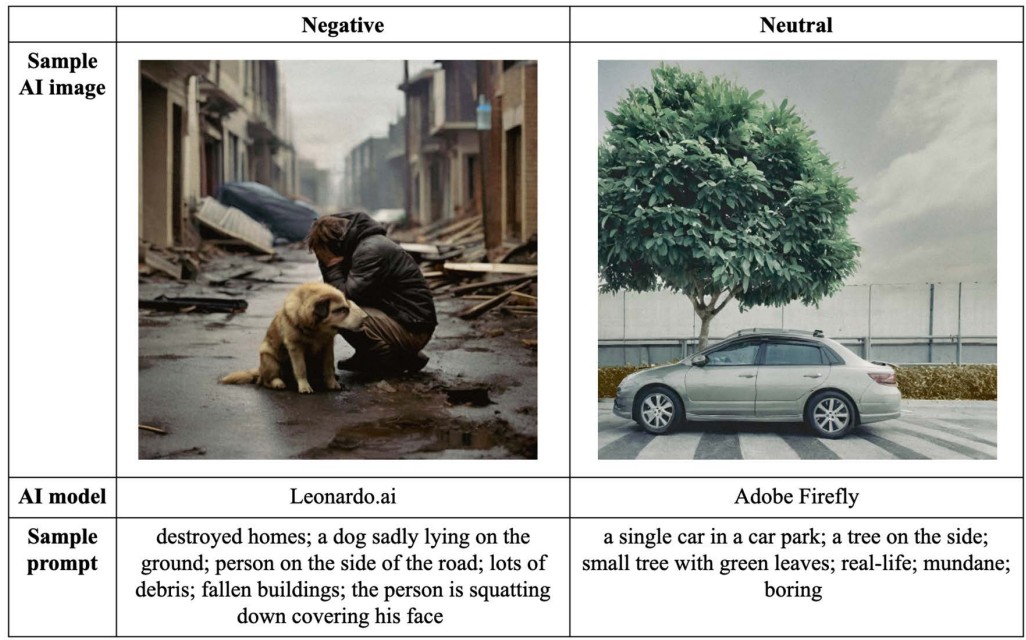

| | Negative | Neutral |
|---|---|---|
| **Sample AI image** | | |
| **AI model** | Leonardo.ai | Adobe Firefly |
| **Sample prompt** | destroyed homes; a dog sadly lying on the ground; person on the side of the road; lots of debris; fallen buildings; the person is squatting down covering his face | a single car in a car park; a tree on the side; small tree with green leaves; real-life; mundane; boring |

**Fig 1. Sample AI-generated images for negative and neutral emotion categories.** Left = people-animal (PA) as a content category. Right = tree-vehicle (TV) as a content category.

the AI tools themselves. For example, Adobe Firefly allowed us to regenerate specific areas of an image with targeted prompts, while in Stable Diffusion, adjusting parameters such as prompt strength enabled greater control over how closely the output adhered to the original prompt.

However, our primary goal was not to create images indistinguishable from real-life photographs but to elicit the intended emotional response (negative or neutral). Consequently, we accepted generated images even when they appeared distinctly "AI-like". For example, some generated images showed backgrounds that lacked detail or appeared blurred compared to actual photographs, while others depicted target objects disproportionate to their backgrounds, or placed in unusual positions. Since these images were presented briefly (around 3–4 seconds) during experimental tasks, neither the level of realism nor participants' recognition of images as AI-generated was considered critical. Following generation, a post-processing workflow was applied. When necessary, we used traditional image editing software (e.g., Adobe Photoshop) to adjust object proportions, placements, and refine color, saturation and brightness. All stimuli were then cropped to 1080 x 1080 pixels and Adobe Lightroom was used to apply a uniform color tone filter across the entire set of 160 images.

**Participants.** 74 participants (64 females, $M_{age}$ = 20.6, $SD$ = 1.91) were recruited using convenience sampling via mass email within the university community. Participants were screened online based on the following inclusion criteria: (1) aged between 18–25, (2) fluent in Cantonese, (3) able to read Traditional Chinese, and (4) normal or corrected-to-normal vision. Given the recruitment method, most participants were undergraduate students (85.1%), followed by postgraduate students (13.5%) and university staff (1.4%). Written informed consent was obtained from all participants. The experimental protocol was approved by the ethics committee of the Chinese University of Hong Kong (SBRE-22–0675).

**Stimulus presentation and rating scales.** The 160 AI-generated images (80 negative and 80 neutral) were divided evenly into two sets of 80. To minimize fatigue, each participant rated the images across two separate runs, with one set assigned to each run. For attentional checks, eight positive images sourced from the Google database were randomly inserted into each set, bringing the total to 88 images per run. These images were pseudo-randomized into four blocks of

22, with constraint that no more than two stimuli from the same emotional valence (negative, neutral or positive) or image category appeared consecutively. The presentation order of blocks was counterbalanced across participants using a Latin square design.

Participants received instructions through a recorded PowerPoint presentation, which explained the 9-point Self-Assessment Manikin (SAM) [13] rating scales for valence and arousal in Cantonese (see Fig 2). These instructions were adapted from the IAPS technical manual. For valence ratings, participants responded to the prompt, "This image is…" on a scale ranging from 1 ("very negative") to 9 ("very positive"). For arousal ratings, participants responded to the prompt, "My reaction to this image is…" on a scale ranging from 1 ("weakly aroused") to 9 ("highly aroused"). Before the main task, participants familiarized themselves with the procedure by completing three practice trials using images not included in the study.

The trial sequence for the image rating task is illustrated in Fig 3. Each trial began with a white fixation cross displayed for 2 seconds to orient participants' attention, followed by the target image displayed for 2 seconds (700 x 700 pixels). Immediately afterward, a smaller version of the same image (500 x 500 pixels) appeared above the valence rating scale. After participants submitted their valence rating, the scale was replaced by the arousal rating scale for the second rating. Both valence and arousal ratings were self-paced and entered using the number keys on the upper-left corner of the keyboard. A 2-second blue fixation cross then appeared, signaling the end of the trial and preparing the participant for the next image. The whole procedure consisted of two runs, with each run comprising four blocks of trials. To minimize fatigue, participants were offered a self-paced break of at least one minute after completing each block. The study was conducted on standard PCs with 24-inch monitors and stimuli were presented using PsychoPy [50].

**Procedure.** Participants visited the laboratory individually or in pairs. They first completed a behavioral task as part of the larger study and were then given the instructions for the image rating task. They were informed that there were no right or wrong answers and were encouraged to provide their honest responses when viewing the images. Upon completion of ratings for the first image set, participants took a mandatory 1-minute break before proceeding to rate the second image set. After completing the entire experiment, each participant received HKD $60 as compensation for their time.

**Statistical analyses.** Descriptive statistics, including mean and standard deviations, were calculated for each image. Inter-rater reliability of the ratings was assessed using intra-class correlation coefficient (ICC). Scatterplots were generated to illustrate the relationships between valence and arousal ratings, allowing visualization of the bidimensional affective space and comparison of the current sample's rating distributions with those of previous studies. Independent samples $t$-tests were conducted to evaluate differences in valence and arousal ratings between neutral and negative stimuli. Pearson's correlation coefficients ($r$) were calculated separately for neutral and negative images to clarify the

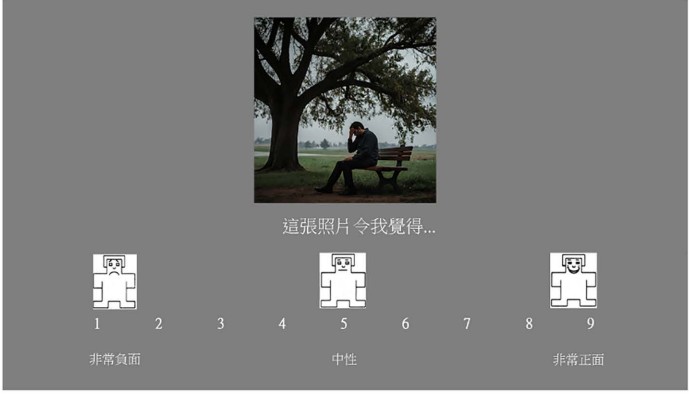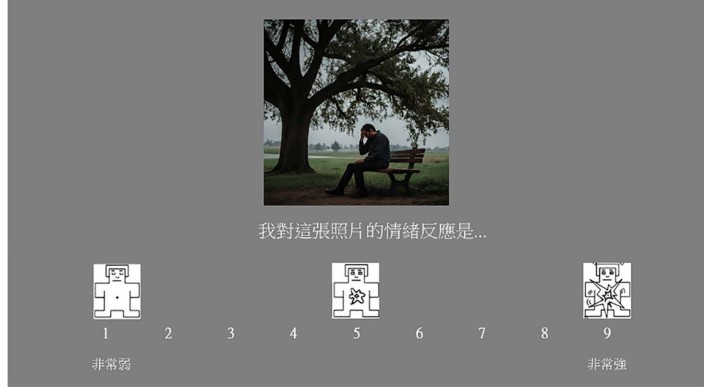

**Fig 2. Display of the SAM scale for Valence and Arousal.** Rating scale presentation. Left = valence, Right = arousal.

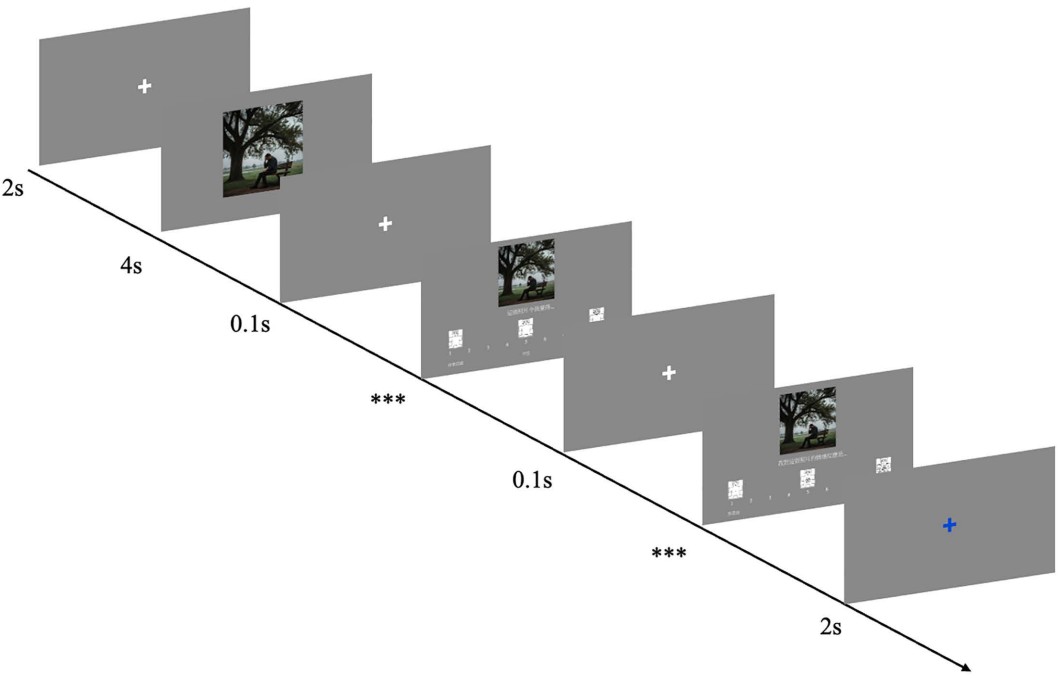

**Fig 3. Trial Sequence of the Rating Procedure.** Ratings were provided for valence first, then arousal. Duration is shown in seconds. *** = self-paced duration.

relationship between valence and arousal ratings. Additionally, linear and quadratic regressions were performed to further investigate valence as a predictor of arousal. All statistical analyses were conducted using SPSS 21 and JASP version 0.18.3 and scatterplots were generated using R Studio.

## Results

This study examined affective ratings for a total of 160 AI-generated images, comprising 80 neutral and 80 negative stimuli. Detailed ratings for all images are provided in S2 Table.

**Data cleaning.** Given the self-paced nature of the rating task, responses with reaction times (RT) shorter than 200ms were removed as such brief response times indicate insufficient evaluation of the stimuli. Additionally, data from seven participants were excluded due to technical errors with the experimental software. Data from two additional participants were excluded because their average rating durations were excessively long (more than 3 SD above the group mean). These exclusions ensured that the analyzed sample had comparable exposure durations to the AI-generated images. Therefore, the final analyzed sample consisted of 65 participants (Female = 57, $M_{age}$ = 20.7, $SD$ = 1.94).

**Rating reliability.** Inter-rater reliability for both valence and arousal ratings was assessed by computing ICC and their 95% confidence intervals, using a two-way mixed-effects model with consistency-agreement for multiple raters (ICC 3, $k$) [51]. The ICC values indicated excellent reliability for both valence (ICC = 0.993, 95% CI [0.991, 0.995]) and arousal (ICC = 0.953, 95% CI [0.942, 0.963]) ratings.

**Rating distribution.** Descriptive statistics for valence and arousal ratings across neutral and negative images are shown in Table 2. For neutral images, the mean valence rating was 5.81 ($SD$ = 0.74, range: 4.02–7.42) and the mean

**Table 2. Descriptive statistics for valence and arousal ratings of images in Study 1 (n = 65).**

| | Negative | | | Neutral | | |
|---|---|---|---|---|---|---|
| | *M* | *SD* | Range | *M* | *SD* | Range |
| Valence | 2.54 | 0.58 | 1.46 - 4.36 | 5.81 | 0.74 | 4.02 - 7.42 |
| Arousal | 5.14 | 0.63 | 3.67 - 6.77 | 3.78 | 0.64 | 2.32 - 2.63 |

*M* and *SD* = mean and standard deviation, respectively

arousal rating was 3.78 (*SD* = 0.64, range: 2.63–4.95). For negative images, the mean valence rating was 2.54 (*SD* = 0.58, range: 1.46–4.36) and the mean arousal rating was 5.14 (*SD* = 0.63, range: 3.67–6.77). Overall, valence ratings ranged from 1.46 to 7.42 indicating that some neutral images were perceived as relatively positive, though their overall mean valence (5.81) remained close to the midpoint of the scale. In contrast, arousal ratings had a narrower range, from 2.32 to 6.77.

**Relationship between valence and arousal.** Independent-sample *t*-tests were conducted to evaluate the differences in valence and arousal ratings between negative and neutral images. Degrees of freedom were adjusted when Levene's test indicated unequal variances. Results revealed significant differences between neutral and negative stimuli in both valence ratings, $t(149) = -30.94$, $p < .001$, 95% CI [−3.47, −3.05] and arousal ratings, $t(158) = 13.56$, $p < .001$, 95% CI [1.16, 1.56]. Both differences indicated medium effect sizes (Cohen's $d = 0.66$ for valence, $d = 0.63$ for arousal). These findings indicate that the AI-generated affective stimuli successfully elicited distinct responses in valence and arousal.

Pearson's correlation coefficients were computed to further examine associations between valence and arousal ratings separately for both negative and neutral images. Among negative images, valence ratings correlated negatively with arousal ($r = -.72$, $p < .001$), indicating that images rated as more negative elicited higher arousal ratings. Conversely, for neutral images, valence ratings correlated positively with arousal ($r = .65$, $p < .001$), indicating that images rated as relatively more positive elicited higher arousal ratings.

Fig 4 shows the scatterplot of arousal versus valence ratings. Specifically, the highest arousal ratings were associated with the lowest valence ratings (the most negative images) and with neutral images rated more positively. Given that our study did not include AI-generated positive stimuli, the upward trend on the positive side of the valence scale was less prominent. To statistically confirm this quadratic relationship, linear and quadratic regression analyses were performed, with mean valence scores and squared mean valence entered as predictors of arousal. Model comparisons indicated that the quadratic regression ($R^2 = 0.769$) provided a substantially better fit than the linear regression model ($R^2 = 0.443$).

## Study 2

### Method

As mentioned above, Study 1 participants completed the image ratings after performing a behavioral task that exposed them to the AI-generated images beforehand. To examine whether image ratings were consistent across independent samples, in addition to ruling out the potential influence of habituation effects, Study 2 recruited a new group of participants who had no prior exposure to the stimuli. Study 2 used a similar design and procedure as Study 1, with the additional set of 40 new AI-generated images that followed the same content combinations described in Study 1. Therefore, Study 2 included a total of 200 images (100 negative, 100 neutral). Table 3 presents the number of images included in each category.

**Participants.** A new sample of 87 participants (49 females; $M_{age} = 20.9$, $SD = 1.89$) was recruited using convenience sampling via university mass mail. Inclusion criteria matched Study 1 exactly: (1) aged between 18–25 years, (2) Cantonese-speaking, (3) able to read Traditional Chinese, and (4) normal or corrected-to-normal vision. Most participants were undergraduate students (83.9%), followed by postgraduate students (12.6%) and university staff

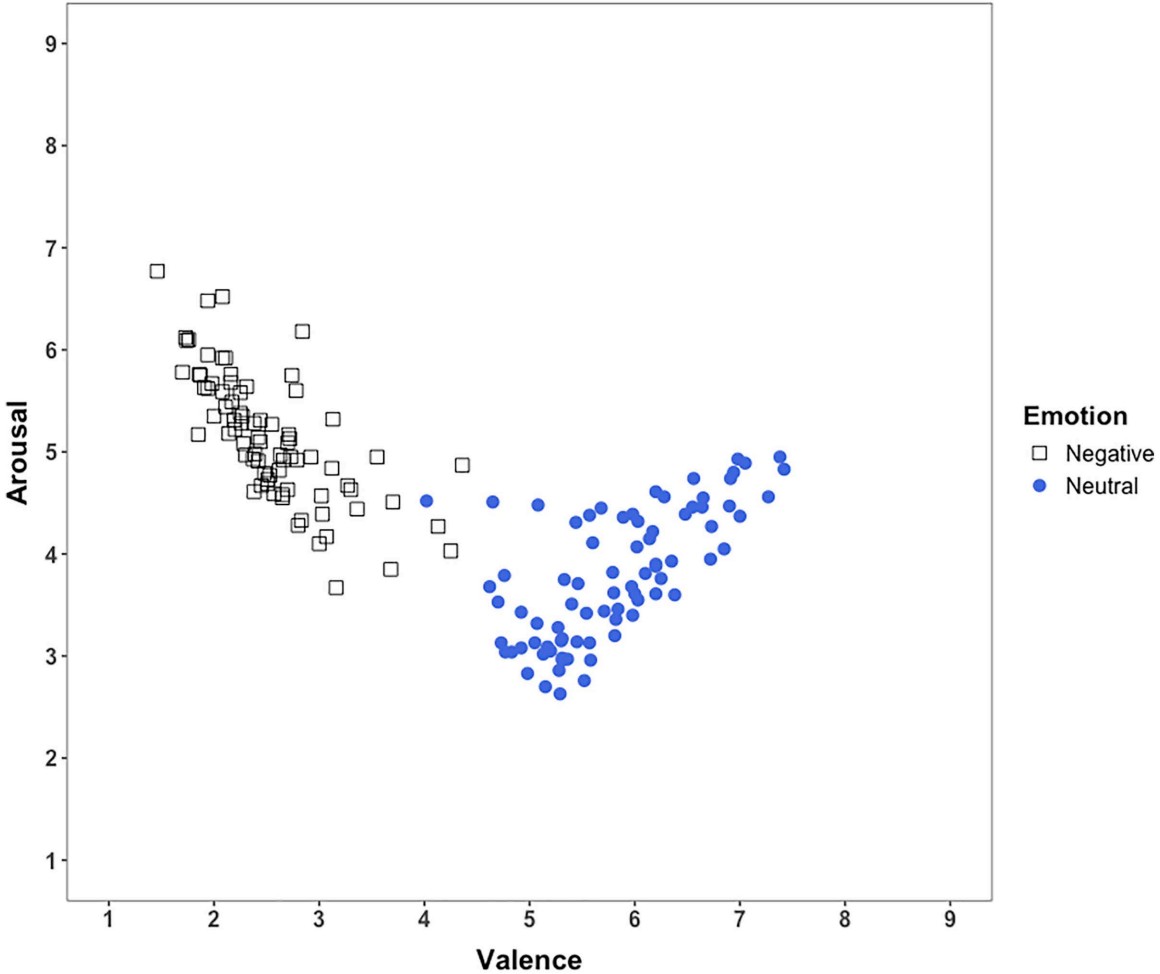

**Fig 4. Scatterplot illustrating the relationship between valence and arousal ratings for the 160 AI-generated images.**

**Table 3. Number of images rated for per category in Study 2.**

| Category | Number of Images | |
|---|---|---|
| | Negative | Neutral |
| PT | 24 | 13 |
| PA | 13 | 24 |
| PV | 13 | 13 |
| TA | 13 | 13 |
| TV | 13 | 24 |
| VA | 24 | 13 |
| Total | 100 | 100 |

Each image contained two content categories as indicated by: PT = people-tree, PA = people-animal, PV = people-vehicle, TA = tree-animal, TV = tree-vehicle, VA = vehicle-animal.

members (3.5%). Three female participants were excluded because they had difficulty understanding the Cantonese instructions during task, despite self-reporting as Cantonese speakers. Hence, the final sample consisted of 84 participants.

To mitigate the risk of non-compliance (e.g., participants providing identical ratings across all) and to reduce potential fatigue from rating images, participants were divided into two groups. Each group rated a subset of the images (Group 1: n = 43; 24 females, $M_{age}$ = 21.1, $SD$ = 1.67; Group 2: n = 41; 22 females, $M_{age}$ = 20.6, $SD$ = 1.84). Written informed consent was obtained from all participants. The experimental protocol was approved by the ethics committee of the Chinese University of Hong Kong (SBRE-22–0675).

**Stimulus presentation and rating scales.** Study 2 included a total of 200 AI-generated images (100 negative and 100 neutral images), these images were divided into two sets (Set 1 and Set 2), each containing 100 images (50 negative and 50 neutral). Group 1 rated the images in Set 1, and Group 2 rated the images in Set 2. Similar to Study 1, eight positive images were randomly interspersed within each set as attentional checks, resulting in 108 images per set. These images were pseudo-randomized into four blocks (27 images per block), ensuring that images from the same emotional valence or content category did not appear more than twice consecutively. Block presentation order was counterbalanced across participants using a Latin square design. Task instructions, practice trials, trial sequence and rating scales were identical to those in Study 1. The experiment concluded after participants completed ratings for all four image blocks. Stimuli were presented on standard PCs with 24-inch monitors using PsychoPy [50].

**Procedure.** The procedure was identical to Study 1, except for two key differences: (1) participants completed the rating task in a single run with 108 images, and (2) participants did not perform any other behavioral tasks prior to image rating. Participants received HKD $60 upon completion as compensation for their time.

**Statistical analyses.** The statistical analyses were identical to those performed in Study 1, conducted separately for each participant group.

## Results

This study examined affective ratings for a total of 200 AI-generated images, comprising 100 neutral and 100 negative stimuli. Detailed ratings for all images are provided in S3 Table.

**Data cleaning.** Similar to Study 1, rating responses with RT shorter than 200ms were excluded, as these indicated inadequate time for proper judgments. Additionally, data from three participants were excluded because their average response times exceeded 3 SD above the group mean. Therefore, the final analyzed sample were N = 43 for Group 1 (24 females, $M_{age}$ = 21.2, $SD$ = 1.68) and N = 38 for Group 2 (22 females, $M_{age}$ = 20.6, $SD$ = 1.84).

**Rating reliability.** Inter-rater reliability was assessed separately for each group by computing ICC and their 95% confidence intervals using a two-way mixed-effects model with consistency agreement for multiple raters (ICC 3, $k$) [51]. For Group 1, ICC values indicated excellent reliability for valence (ICC = 0.988, 95% CI [0.984, 0.991]) and arousal (ICC = 0.935, 95% CI [0.916, 0.952]). Similarly, Group 2 demonstrated excellent reliability for valence (ICC = 0.989, 95% CI [0.985, 0.992]) and arousal (ICC = 0.960, 95% CI [0.948, 0.970]).

**Rating distribution.** Tables 4 and 5 show descriptive statistics for valence and arousal ratings collected from Group 1 and 2, respectively. For Group 1, negative images had a mean valence rating of 2.73 ($SD$ = 0.60, range: 1.55–4.49) and a mean arousal rating of 5.10 ($SD$ = 0.75, range: 3.80–7.12). Neutral images had a mean valence rating of 5.71 ($SD$ = 0.66, range: 4.56–7.12) and mean arousal rating of 3.65 ($SD$ = 0.61, range: 2.54–5.08).

For Group 2, negative images had a mean valence rating of 2.65 ($SD$ = 0.56, range: 1.76–4.16) and mean arousal rating of 5.24 ($SD$ = 0.85, range: 3.50–7.18). The neutral images had a mean valence rating of 5.68 ($SD$ = 0.81, range: 3.61–7.76) and mean arousal rating of 3.16 ($SD$ = 0.76, range: 2.09–4.74).

**Table 4. Descriptive statistics for valence and arousal ratings of images evaluated by Group 1 (n = 43) in Study 2.**

|  | Negative | | | Neutral | | |
|---|---|---|---|---|---|---|
|  | **M** | **SD** | **Range** | **M** | **SD** | **Range** |
| Valence | 2.73 | 0.60 | 1.55 - 4.49 | 5.71 | 0.66 | 4.56 - 7.12 |
| Arousal | 5.10 | 0.75 | 3.80 - 7.12 | 3.65 | 0.61 | 2.54 - 5.08 |

*M* and *SD* = mean and standard deviation, respectively.

**Table 5. Descriptive statistics for valence and arousal ratings of images evaluated by Group 2 (n = 38) in Study 2.**

|  | Negative | | | Neutral | | |
|---|---|---|---|---|---|---|
|  | **M** | **SD** | **Range** | **M** | **SD** | **Range** |
| Valence | 2.65 | 0.56 | 1.76 - 4.16 | 5.68 | 0.81 | 3.61 - 7.76 |
| Arousal | 5.24 | 0.85 | 3.50 - 7.18 | 3.16 | 0.76 | 2.09 - 4.74 |

*M* and *SD* = mean and standard deviation, respectively.

Overall, valence ratings from both groups indicated that some neutral images received ratings toward the positive end of the scale (Group 1 max = 7.12, Group 2 max = 7.76). However, the mean valence for neutral images (Group 1, $M$ = 5.71, Group 2, $M$ = 5.68) remained near the midpoint of the scale. Arousal ratings showed somewhat narrower ranges, spanning from 2.54 to 7.12 for Group 1 and from 2.09 to 7.18 for Group 2.

**Relationship between valence and arousal.** Independent-sample t-tests were conducted to evaluate the differences in valence and arousal ratings between negative and neutral images. Degrees of freedom were adjusted when Levene's test indicated unequal variances. Results revealed significant differences between negative and neutral images for both valence ratings (Group 1: $t(98)$ = −23.60, $p < .001$, 95% CI [−3.22, −2.72], Cohen's $d$ = 0.63; Group 2: $t(87.6)$ = −21.74, $p < .001$, 95% CI [−3.30, −2.75], Cohen's $d$ = 0.70) and arousal ratings (Group 1: $t(98)$ = 10.59, $p < .001$, 95% CI [1.18, 1.72], Cohen's $d$ = 0.68; Group 2: $t(98)$ = 12.94, $p < .001$, 95% CI [1.76, 2.40], Cohen's $d$ = 0.80). Effect sizes of these comparisons ranged from medium to large. Thus, these results indicate that the AI-generated affective stimuli successfully elicited distinct differences in valence and arousal ratings across two independent participant samples.

Pearson's correlations were calculated separately for negative and neutral images within each group. For Group 1, neutral images showed a significant positive correlation between valence and arousal ratings ($r$ = .64, $p < .001$), indicating that neutral images perceived as more positive were also more arousing. Conversely, negative images showed a significant negative correlation between valence and arousal ratings ($r$ = −.70, $p < .001$), indicating that images perceived as more negative were more arousing.

Group 2 revealed a similar pattern: neutral images showed a significant positive correlation between valence and arousal ratings ($r$ = .53, $p < .001$), indicating that neutral images perceived as more positive were also more arousing. Negative images showed a strong negative correlation ($r$ = −.87, $p < .001$), suggesting that more negative images were perceived as more arousing.

Scatterplots depicting the relationship between valence and arousal ratings for both groups are presented in Fig 5. As expected, both groups demonstrated highest arousal ratings for images at the lowest valence end (most negative images) and at the neutral images rated toward the positive end of the valence scale.

To statistically confirm the quadratic relationship between valence and arousal, linear and quadratic regression analyses were performed separately for each group, with mean valence and squared mean valence entered as predictors of arousal. Results indicated a better model fit for quadratic regression compared to linear regression

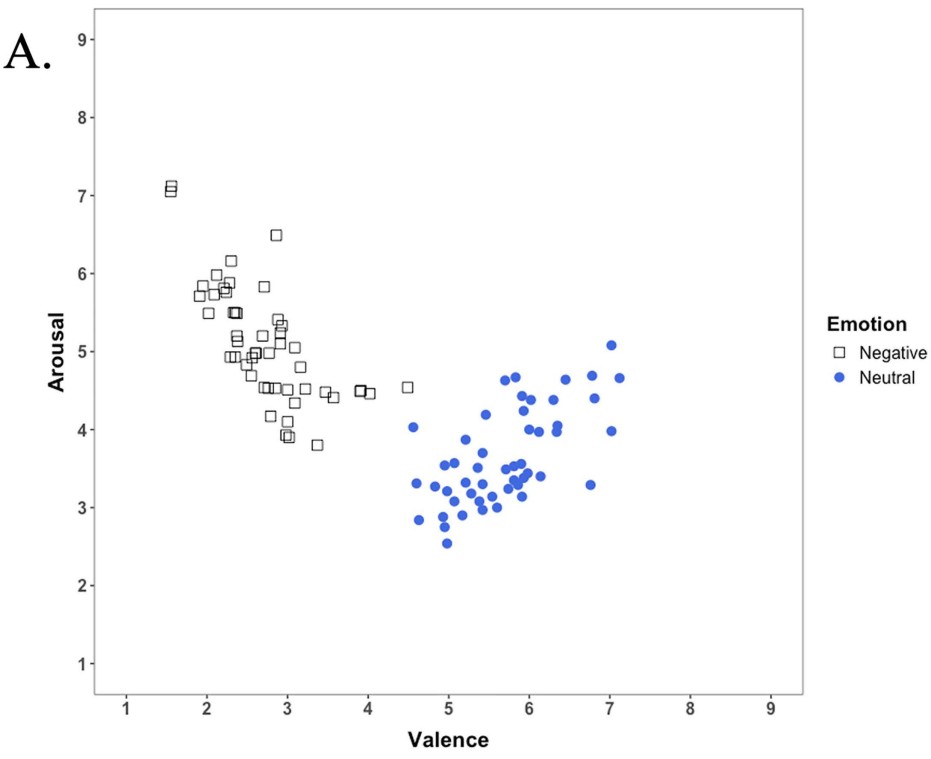

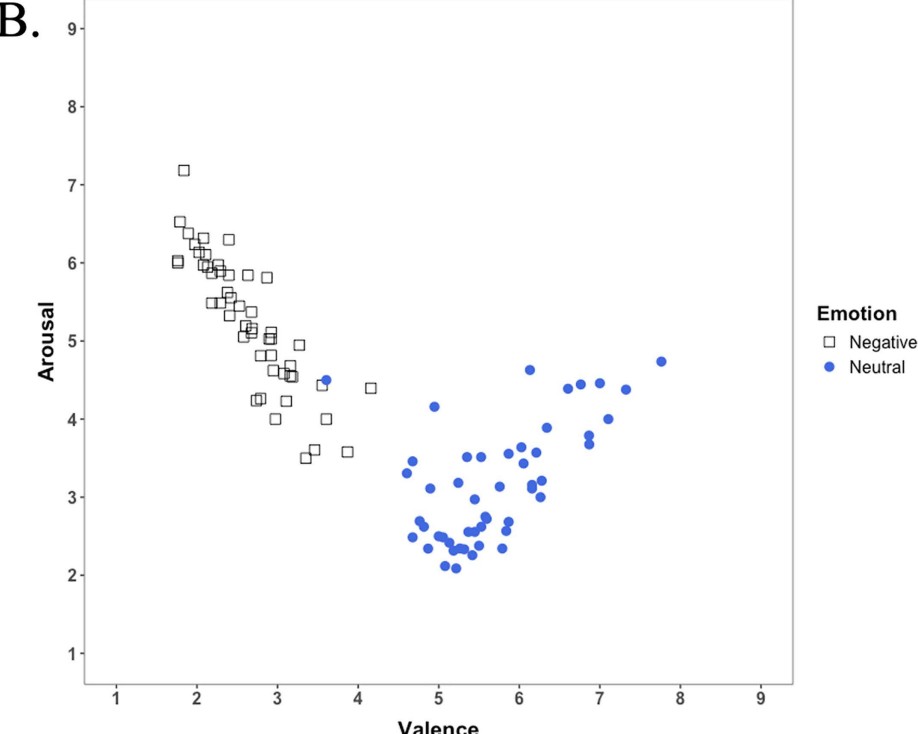

**Fig 5. Scatterplots illustrating the relationship between valence and arousal ratings for the 200 AI-generated images.** Each group rated a separate set of 100 images (50 negative and 50 neutral). **(A)** Ratings from Group 1 (n = 43); **(B)** Ratings from Group 2 (n = 38).

in both groups: Group 1 (quadratic $R^2 = 0.748$ vs. linear $R^2 = 0.477$) and Group 2 (quadratic $R^2 = 0.865$ vs. linear $R^2 = 0.552$).

## Discussion

This study aimed to assess the feasibility and effectiveness of using AI-generated negative and neutral naturalistic scene stimuli in affective research. Using three text-to-image AI generation models, we developed a set of 200 images (100 negative and 100 neutral), carefully controlling the combination of two content categories to meet the specific experimental criteria. We collected valence and arousal ratings from multiple participant samples to explore three key questions: (1) whether AI-generated images effectively evoke emotional responses, (2) whether these images demonstrate similar affective rating patterns as previously validated standardized databases and (3) whether AI image generation can be tailored effectively to specific experimental designs. We discuss these points below.

The findings from both Study 1 and Study 2 indicate that AI-generated negative and neutral images elicited distinct ratings for valence and arousal. Consistently across both studies and all participant samples, negative images were rated significantly lower in valence (more negative) and higher in arousal compared to neutral images. Moreover, valence and arousal ratings exhibited differing correlations depending on image category: negative images showed a negative correlation (lower valence associated with higher arousal), whereas neutral images demonstrated a positive correlation (higher valence associated with higher arousal). Despite the potential influence of habituation in Study 1, the consistency of the results from Study 1 and Study 2 demonstrates the robustness of the affective images generated by AI. Importantly, the relationship between valence and arousal ratings in both studies conformed to the expected quadratic distribution, specifically within the range of negative to neutral valence. These findings align closely with previously reported patterns for negative and neutral stimuli from normative affective image databases (e.g., [2,10,11]) and in rating studies using standardized stimuli (e.g., [1]).

The mean valence and arousal ratings of our AI-generated negative and neutral images were comparable to those reported in existing affective databases (e.g., [9,37,52]). Specifically, across our three participant samples, mean valence ratings for negative images ranged from 2.54 to 2.73, and mean arousal ratings ranged from 5.10 to 5.24. Neutral images generally clustered around the midpoint of the valence scale; however, some neutral images received higher-than-expected valence ratings (up to 7.76 in Study 2). We suspect that these evaluated ratings may reflect participants' relative judgment influenced by the substantial proportion of negative stimuli, while positive images (those for attentional checks) were very limited. Participants were broadly informed that images could elicit various emotions, which may have encouraged them to utilize the full range of the SAM rating scales, resulting in some neutral images being perceived as slightly positive. Nevertheless, the overall mean valence and arousal ratings of the negative and neutral images closely align with normative data from established affective databases [10,37]. These findings support and extend recent research demonstrating that AI-generated stimuli can indeed elicit distinct emotional responses (e.g., [28,36]).

Our findings suggest that AI text-to-image generators can be effectively employed to develop stimuli that precisely match specific experimental requirements and content constraints. In the present study, we targeted six combinations of relatively common content categories (animals, people, vehicle and trees). However, it remains unclear whether AI generators *alone* can reliably produce large and sufficiently varied sets of images depicting highly specialized or emotionally explicit content (e.g., war, violence) that may be necessary for certain research contexts (e.g., PTSD in war veterans). Additionally, some freely accessible AI platforms restrict generation of explicit negative content, such as violence, gore, or abuse. While offline or specialized AI models could potentially bypass such limitations, the extent to which these models can generate varied and usable content remains an area requiring further exploration.

While AI-generated images can be created rapidly (often in less than a minute), the initial outputs frequently require refinement, as generated images might not always align perfectly with intended emotional or content specifications. Iterative refinement of text prompts or manual editing using conventional image-editing software can be time-consuming. Moreover,

reproducibility of the image using an identical prompt is difficult with free generative AI platforms, as there is minimal control over generation parameters or random seed values. In other words, if one uses the same prompt repeatedly, the resulting image would not be identical. We acknowledge that this is not ideal if researchers wish to share their text prompts with others to produce the same images. However, the text prompts should still be useful in the generation of similar themed images.

As mentioned earlier, our AI-generated images were intended solely to elicit emotion in standard affective research tasks. Considering that the presentation times are often fairly brief (e.g., a few seconds as an emotional distractor), we did not extensively edit the images to enhance realism. With the rapid advancement of AI-generation models, we expect that the translation of the prompts to the generated image will become increasingly accurate and show significant improvements in realism over time, reducing the efforts required for additional manual editing. Therefore, despite certain disadvantages in reproducibility and the realism of generated outputs, AI text-to-image generators remain an attractive alternative for researchers seeking novel or additional stimuli, particularly when existing databases lack suitable content or when new stimuli are needed to mitigate habituation effects.

Several limitations of the current study should be noted. First, due to practical constraints, positive images were not generated or rated. Consequently, our evaluation was restricted to the negative and neutral spectrum, preventing us from determining whether AI-generated positive images would replicate the typical U-shaped distribution observed in traditional affective databases. Second, we did not systematically assess whether participants were aware of the AI-generated nature of the stimuli. Although we lacked a formal protocol, anecdotal reports indicated that a minority of participants (Study 1: $n = 11$; Study 2: $n = 19$) identified the images as AI-generated. We have included a descriptive comparison of valence and arousal ratings between these participants and the remaining sample in the Supporting Information (see S4 Table). While these groups appear to show similar ratings, we refrained from formal statistical analysis due to the unsystematic data collection and small sample size of the "aware" group. The question of how knowledge of AI origins influences emotional response remains an open area of inquiry. Recent literature suggests that while awareness of AI origins may create an implicit bias, such as altered fixation durations [53] or physiological responses [54], these effects do not necessarily extend to explicit subjective ratings of valence and arousal. Although our preliminary observations align with this literature, suggesting that subjective ratings remain robust despite AI awareness, future research should systematically investigate how explicit knowledge of image origin influences emotional processing.

It is important to note that AI-generated images are not intended to replace the value of standardized affective stimuli databases, which provide large-scale normative rating data. Rather, they serve as a powerful complementary tool. Because normative data for AI-generated stimuli are not yet widely available, we recommend that researchers conduct pilot ratings with an independent sample and collect ratings from the experimental sample itself to confirm that the stimuli elicit the intended emotional responses.

In summary, this study demonstrates the substantial potential of AI text-to-image generation for stimulus development in affective research. Our findings provide an initial validation, indicating that AI-generated negative and neutral images elicit emotional responses and exhibit valence-arousal patterns that closely resemble those from standardized databases. As generative AI technologies advance, they will likely facilitate the efficient creation of tailored, high-quality stimuli not only in visual domains but also across audio, video and textual modalities. Ultimately, the adoption of AI-generated stimuli can substantially streamline stimuli development, reduce the burden of stimulus selection and enhance methodological flexibility in affective research.

## Supporting information

**S1 Table. Example prompts and generated outputs from Adobe Firefly, Stable Diffusion and Leonardo.ai.** Example prompts were used in the three AI platforms in early 2024. Newer AI-generative models will produce different outputs when using the above parameters and text prompts.
(DOCX)

**S2 Table. Valence and arousal ratings of images in Study 1 (n = 65).** PT = people-tree, PA = people-animal, PV = people-vehicle, TA = tree-animal, TV = tree-vehicle, VA = vehicle-animal.
(XLSX)

**S3 Table. Valence and arousal ratings of images in Study 2 (set 1, n = 43; set 2, n = 38).** PT = people-tree, PA = people-animal, PV = people-vehicle, TA = tree-animal, TV = tree-vehicle, VA = vehicle-animal.
(XLSX)

**S4 Table. Descriptive statistics for valence and arousal ratings for participants who noticed the AI-generated images ("Aware") versus those who did not ("Unaware") in Study 1 and Study 2.** *M* and *SD* = mean and standard deviation, respectively.
(DOCX)

## Author contributions

**Conceptualization:** Hey Tou Chiu, Savio W.H. Wong.

**Data curation:** Yuen Wing Lam.

**Formal analysis:** Hey Tou Chiu, Hoi In Sou, Yuen Wing Lam, Clayton Siu Fung Ng.

**Funding acquisition:** Savio W.H. Wong.

**Investigation:** Hey Tou Chiu, Hoi In Sou, Yuen Wing Lam, Clayton Siu Fung Ng.

**Methodology:** Hey Tou Chiu, Yuen Wing Lam, Clayton Siu Fung Ng, Savio W.H. Wong.

**Project administration:** Hey Tou Chiu, Hoi In Sou.

**Resources:** Savio W.H. Wong.

**Supervision:** Hey Tou Chiu, Savio W.H. Wong.

**Writing – original draft:** Hoi In Sou.

**Writing – review & editing:** Hey Tou Chiu, Yuen Wing Lam, Clayton Siu Fung Ng, Savio W.H. Wong.

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
