## [Decision Letter · Decision Letter 0]

18 Aug 2025

Dear Dr. Wong,

Thank you for submitting your manuscript to PLOS ONE. After careful consideration, we feel that it has merit but does not fully meet PLOS ONE’s publication criteria as it currently stands. Therefore, we invite you to submit a revised version of the manuscript that addresses the points raised during the review process.

We look forward to receiving your revised manuscript.

Kind regards,

Sihua Xu

Academic Editor

PLOS ONE

Journal Requirements:

[General Research Fund from the Research Grant Council Hong Kong (RGC Ref No. 14619919)

Social Innovation and Entrepreneurship Development Fund (SIE Fund) KPF20QEP12.].

3. We note that Figures 1, 2, and 3 in your submission may contain copyrighted images. All PLOS content is published under the Creative Commons Attribution License (CC BY 4.0), which means that the manuscript, images, and Supporting Information files will be freely available online, and any third party is permitted to access, download, copy, distribute, and use these materials in any way, even commercially, with proper attribution. For more information, see our copyright guidelines: http://journals.plos.org/plosone/s/licenses-and-copyright.

1. You may seek permission from the original copyright holder of Figures 1, 2, and 3 to publish the content specifically under the CC BY 4.0 license.

Additional Editor Comments:

Dear Dr. Wong,

Thank you for submitting your manuscript entitled “Beyond Traditional Stimuli: Establishing AI-Generated Images as Valid Tools for Emotion and Affect Research” (Manuscript ID: PONE-D-25-25040) to PLOS ONE. Your work addresses a timely and novel question regarding the validity of AI-generated images as affective stimuli. Reviewers recognized several strengths, including the originality of the topic, appropriate sample sizes and statistical analyses, high inter-rater reliability, and transparent reporting of unexpected findings.

At the same time, all reviewers identified substantial issues that need to be resolved before the manuscript can be considered further. Key concerns include: the need for extensive English language editing and clearer presentation; missing or incomplete supplementary materials; insufficient articulation of the study’s purpose, hypotheses, novelty, and relation to prior literature; and the unclear necessity of Study 1 given habituation confounds. Reviewers also emphasized that the complete omission of positive stimuli substantially limits the study’s claims, and that issues of ecological validity, participant awareness of AI-generated images, and the reproducibility of stimuli (including disclosure of prompts and model parameters) require fuller justification and discussion.

In light of these points, the editorial decision is Major Revision. We invite you to submit a thoroughly revised manuscript that addresses the reviewers’ comments in detail. Careful attention to these issues will be essential to strengthen the contribution and ensure the manuscript meets the standards of PLOS ONE.

Reviewers' comments:

Reviewer's Responses to Questions

**Comments to the Author**

1. Is the manuscript technically sound, and do the data support the conclusions?

Reviewer #1: Yes

Reviewer #2: Partly

Reviewer #3: Partly

2. Has the statistical analysis been performed appropriately and rigorously?

Reviewer #1: Yes

Reviewer #2: Yes

Reviewer #3: Yes

3. Have the authors made all data underlying the findings in their manuscript fully available?

Reviewer #1: Yes

Reviewer #2: Yes

Reviewer #3: No

4. Is the manuscript presented in an intelligible fashion and written in standard English?

Reviewer #1: No

Reviewer #2: Yes

Reviewer #3: Yes

Reviewer #1: Recommendations for Manuscript ID PONE-D-25-25040 Title: “ Beyond Traditional Stimuli: Establishing AI-Generated Images as Valid Tools for Emotion and Affect Research” for the Plos One Journal.

General Comments

From my point of view, it is a very interesting topic and simultaneously it seems that to the best of my knowledge is an empirical research aims to explored the feasibility of using generative AI, specifically text-to-image generators, to create tailored affective stimuli. Across two studies, participants rated the valence and arousal of 160 and 200 AI-generated images (negative and neutral). Our findings revealed that AI-generated images exhibit the typical valence-arousal patterns observed in standardized affective databases, demonstrating moderate to strong associations between these two emotional dimensions. These results highlight the potential of generative AI as a valuable methodological tool for creating customized affective stimuli aligned with distinct research objectives and experimental designs.

The paper contains the following sections: Introduction, Current uses of artificial intelligence in stimuli development, Present study, Study 1, Study 2, Discussion, .

However, I find some recommendations:

1. The Manuscript needs careful English proofreading because there are some shortcomings. For instance, the article “the” is sometimes missing in front of nouns, the message in some paragraphs is not clear enough. It looks like the first part was written by one author with a greater command of the English language, and the rest of the paper was written by someone else. The numerous grammar errors made this a difficult paper to read. It was strange to see the authors refer to tables that were not submitted. I was unable to find any supplementary material to the submission, so I think this was truly omitted by the authors. Please read the manuscript carefully.

2. The abstract must contain the main purpose of the paper, the research method used in the research and the main contributions.

3. It would be very useful to add in the "Introduction" section the purpose, objectives and hypothesis of the research. I consider that a weak point of the paper is that the authors did not show the novelty of the paper compared to other works. That is why, I consider that the introduction should specify the novelty of the paper compared to other papers published in this area.

4. The Literature Review cannot be missing from the paper.

5. The research is well based on science and the results are in agreement with the theoretical part.

6. I believe that the authors should also include other indicators from Descriptive Statistics.

7. I think that the literature needs to be improved with other recent works, refers to the companies listed on the economic growth. That is why I recommend the authors to refer to other recent works indexed in Web of Science, Scopus, Emerald and Cambrige Journals. We suggest that the authors cite papers published in Web of Science Journals such as:

a. https://doi.org/10.3390/cancers15030843

b. https://doi.org/10.3390/s23052398

c. https://doi.org/10.3390/math11102305

d. https://doi.org/10.3390/fractalfract8100604

In conclusion, the article should be improve. It should also be enhanced with a review of the literature adequate to the subject and a broader interpretation and commentary of the research results.

Reviewer #2: Thank you for this interesting manuscript. Through my reading, I found several strengths of this study as well as its key limitations. In what follows, I will discuss the strengths first and then move on to the key limitations.

The most salient strength of this study pertains to its topic. The authors address a truly novel topic: whether AI-generated images can be valid stimuli for emotion and affect research. I believe that this topic is timely and worthwhile, given the increasing use of AI in academic research.

Another strength I’d like to mention is that the manuscript is well-written and effectively organized. The introduction and literature review sections are smoothly connected. The previous studies mentioned in this early part are highly relevant to the current study and effectively justify why this study is necessary. One important virtue of the manuscript is its clarity. The section “Research gap” is a case in point; the section title is straightforward, and the corresponding content is also easy to follow.

The last strength I appreciate in this manuscript is the authors’ transparency. They openly share even some information that can appear to be against the study’s credibility. For instance, the authors did not hide the fact that neutral images received higher-than-expected valence ratings. While this could have given some doubt to the validity of the stimuli, the authors smartly explained their best guess about the reason, which sounds reasonable and well thought out. I believe that the authors show a good practice of transparency throughout the manuscript.

Despite these strengths, I also found some areas for improvement, which might diminish the study’s value, as follows:

First, it is not clear why the authors have two sub-studies. Study 1 and Study 2 overlap, and it seems obvious that Study 2 is more rigorous than Study 1. The authors stated that the participants in Study 1 saw the stimulus images before Study 1 as part of another study. This is a very significant limitation, which I think the authors cannot get away with. As mentioned in the manuscript as well, habituation effects likely happened. Since the authors did not explain what kind of study the previous study was and in what context the images were displayed, it is difficult to guess how seriously habituation effects occurred there. But, one clear thing is that the previous experience must have harmed the quality of Study 1 to a certain extent. My best guess about why the authors included Study 1 is probably that the consistent findings across the two sub-studies can demonstrate the reliability of their findings. However, I would say that is not the case here, due to the critical limitation of Study 1. Therefore, I suggest the authors retain only Study 2.

Another key limitation of this study is that the study did not test positive images. Although the authors clearly admit this as their limitation in the Discussion section, I do not think this is a simple limitation that they can admit and move on. In fact, not examining positive images makes this study’s findings look half-baked. The true utility of AI-generated images as tools for valence/emotion research can be only confirmed after testing positive images (and positive valences). If there was an unavoidable reason behind this critical exclusion, the authors should clarify it in the manuscript. Without that, the current version leaves some doubt about the completeness of the study.

The two areas for improvement I mentioned above may give an impression in common that the study was designed based on the authors’ convenience rather than scientific criteria. I think it would be ideal if the authors could resolve this potential issue through their revision. I truly hope my comments help and wish the authors all the best in their research. Thank you.

Reviewer #3: The study aims to validate AI-generated images (created via text-to-image prompts) as viable stimuli in emotion research. It is, to the authors' knowledge, the first to systematically assess whether such images can elicit valence–arousal patterns comparable to those found in standardized affective databases such as IAPS. This represents a meaningful methodological contribution to affective psychology, particularly given the increasing interest in customizable and scalable stimulus creation using AI.

The two studies are well-designed, with adequate sample sizes, strong inter-rater reliability (ICC > .95), and the use of validated tools such as the Self-Assessment Manikin (SAM). The statistical analyses—t-tests, correlations, and regression models—are appropriate and competently executed. The results confirm that AI-generated images, at least for negative and neutral content, follow the classic U-shaped valence–arousal curve observed in traditional databases. This finding suggests that such images may offer a flexible alternative for generating emotionally calibrated stimuli in experimental paradigms.

Despite these strengths, the study has several serious limitations that must be addressed before it can be considered for publication.Most notably, the study excludes positive stimuli entirely, undermining the central claim that AI-generated images can replicate the full range of affective responses typically captured in standardized image sets. Without positive images, the U-shaped valence–arousal distribution is only partially validated, which significantly limits the generalizability of the findings. The authors must clearly justify this omission and discuss its implications. Second, no data were collected on whether participants recognized the images as AI-generated. This omission is problematic, as prior work suggests that perceived artificiality can modulate emotional responses. Furthermore, the authors report that images with visual distortions or unrealistic elements were intentionally retained. While this decision may reflect practical constraints, it raises concerns about ecological validity and the applicability of findings to real-world emotional processing.

While the authors briefly acknowledge the absence of positive images and the choice to include AI-distorted outputs, these decisions require more detailed justification and discussion. Specifically, the complete omission of positive stimuli significantly limits the ability to validate the full valence–arousal spectrum and undermines the claim that AI-generated images replicate the distributional patterns observed in standardized databases like IAPS. The rationale for this exclusion is not sufficiently developed. Additionally, although the authors note that realism was not a primary goal, they do not assess whether participants were aware that the images were AI-generated or whether such awareness might have influenced their emotional ratings. Given growing evidence that perceived authenticity can modulate affective responses, this represents a potential confound that should be directly addressed. Both points—the exclusion of positive stimuli and the lack of participant awareness checks—require clearer theoretical and methodological justification.

The paper provides only a few examples of the generated images and their corresponding prompts, which raises concerns about transparency and reproducibility. Given that the study’s main contribution lies in the use of generative AI for flexible stimulus creation, it is essential to disclose a representative and sufficiently detailed sample of prompts, along with generated outputs. Without this information, it is difficult to assess the thematic accuracy, variability, or potential biases in the generation process. Most critically from a methodological standpoint, the authors do not address the inherent stochasticity of text-to-image generation models. Most generative AI systems sample from a probability distribution, meaning that identical prompts can yield different images across runs—especially on platforms like Adobe Firefly and DALL·E, which do not allow for control over random seed values or generation parameters. This severely compromises the reproducibility of the stimuli. The authors’ proposal that AI-generated images could replace standardized image databases is therefore premature unless procedures are put in place to ensure deterministic, shareable outputs. At a minimum, the authors should document the AI model configuration and any image metadata. If the same prompt was used in both Study 1 and Study 2, it would be valuable to examine and report whether the emotional ratings of the resulting images differed—highlighting the variability introduced by prompt reuse without seed control.

**Do you want your identity to be public for this peer review?** For information about this choice, including consent withdrawal, please see our Privacy Policy

Reviewer #1: No

Reviewer #2: No

Reviewer #3: No

---

## [Author Response · Author response to Decision Letter 1]

3 Oct 2025

We greatly appreciate the reviewer’s comments on the manuscript. Below, we have organized the comments that require revision and our point-by-point response to each comment.

Reviewer #1:

1. The Manuscript needs careful English proofreading because there are some shortcomings. For instance, the article “the” is sometimes missing in front of nouns, the message in some paragraphs is not clear enough. It looks like the first part was written by one author with a greater command of the English language, and the rest of the paper was written by someone else. The numerous grammar errors made this a difficult paper to read. It was strange to see the authors refer to tables that were not submitted. I was unable to find any supplementary material to the submission, so I think this was truly omitted by the authors. Please read the manuscript carefully.

- Table 1-5 are enclosed within the main text of the PDF file, which can be viewed using Acrobat Reader. Table S1 and S2 are provided as supporting information and can be accessed by clicking the download links at the end of the PDF file (pages 53 and 54). After accepting the download request, the supporting information in Word document format should be automatically downloaded to the default download folder of the local computer. While other reviewers have not reported a similar issue with accessing the supporting information, we suspect that the problem might be due to different PDF readers. Therefore, we recommend using Acrobat Reader for viewing the PDF file. Additionally, we have carefully proofread the manuscript and made necessary changes to enhance its readability.

2. The abstract must contain the main purpose of the paper, the research method used in the research and the main contributions.

- The abstract has been revised accordingly to clarify the aims and scope of this paper.

3. It would be very useful to add in the "Introduction" section the purpose, objectives and hypothesis of the research. I consider that a weak point of the paper is that the authors did not show the novelty of the paper compared to other works. That is why, I consider that the introduction should specify the novelty of the paper compared to other papers published in this area.

- We have expanded on our literature review on current uses of AI in stimuli development by including more recent studies of AI-generation of facial stimuli (lines 84-89). We have also clarified in the Research Gap and Present Study section that the novelty of this study is to systematically examine the affective dimensions of AI-generated naturalistic scene stimuli (lines 111-119), with the aim of demonstrating that the normative ratings are replicable across independent samples (lines 130-139)

4. The Literature Review cannot be missing from the paper.

- Although there is no dedicated subheading for the literature review, it is incorporated into the introduction section.

5. The research is well based on science and the results are in agreement with the theoretical part.

- We appreciate the positive comments from the reviewer.

6. I believe that the authors should also include other indicators from Descriptive Statistics.

- Descriptive statistics are presented in the main text and Table 2 and 4.

7. I think that the literature needs to be improved with other recent works, refers to the companies listed on the economic growth. That is why I recommend the authors to refer to other recent works indexed in Web of Science, Scopus, Emerald and Cambrige Journals. We suggest that the authors cite papers published in Web of Science Journals such as:

a. https://doi.org/10.3390/cancers15030843

b. https://doi.org/10.3390/s23052398

c. https://doi.org/10.3390/math11102305

d. https://doi.org/10.3390/fractalfract8100604

In conclusion, the article should be improve. It should also be enhanced with a review of the literature adequate to the subject and a broader interpretation and commentary of the research results.

- We appreciate the reviewer’s suggestions and carefully examined the four recommended papers from a pioneering research group that applies advanced mathematical and computational models to complex classification and detection problems across diverse scientific domains. While our study focuses on using existing AI tools to generate negative and neutral images for affective research, we found limited conceptual or methodological overlap with these works. Accordingly, we did not include citations to these papers in the manuscript.

Reviewer #2:

1. First, it is not clear why the authors have two sub-studies. Study 1 and Study 2 overlap, and it seems obvious that Study 2 is more rigorous than Study 1. The authors stated that the participants in Study 1 saw the stimulus images before Study 1 as part of another study. This is a very significant limitation, which I think the authors cannot get away with. As mentioned in the manuscript as well, habituation effects likely happened. Since the authors did not explain what kind of study the previous study was and in what context the images were displayed, it is difficult to guess how seriously habituation effects occurred there. But, one clear thing is that the previous experience must have harmed the quality of Study 1 to a certain extent. My best guess about why the authors included Study 1 is probably that the consistent findings across the two sub-studies can demonstrate the reliability of their findings. However, I would say that is not the case here, due to the critical limitation of Study 1. Therefore, I suggest the authors retain only Study 2.

- We thank the reviewer for this comment and have revised the manuscript (lines 152-165) to clarify the distinct and complementary roles of Study 1 and Study 2.

A primary objective of this paper is to demonstrate the feasibility of using generative AI to create stimuli for affective research. Study 1 was designed to reflect a common practice in the field, where affective ratings are collected from participants after the main experiment. The findings confirmed that the AI-generated negative and neutral stimuli elicited different valence and arousal ratings. However, this design introduced a potential confound of habituation, as the ratings were collected upon the participants' second exposure to the images.

To address this limitation and establish the reliability of our findings, we conducted Study 2. This study provided a methodologically clean validation by collecting ratings upon the first exposure to the stimuli, thereby removing any potential influence of habituation.

By presenting both studies, we demonstrate that the AI-generated images exhibit consistent affective characteristics, regardless of whether ratings are collected during the first or second exposure. This dual-study approach provides a robust validation of the stimuli and strengthens the basis for our recommendation in the discussion: researchers who generate their own stimuli should collect normative ratings after the main experiment if collecting them from a separate, naive sample is not feasible.

2. Another key limitation of this study is that the study did not test positive images. Although the authors clearly admit this as their limitation in the Discussion section, I do not think this is a simple limitation that they can admit and move on. In fact, not examining positive images makes this study’s findings look half-baked. The true utility of AI-generated images as tools for valence/emotion research can be only confirmed after testing positive images (and positive valences). If there was an unavoidable reason behind this critical exclusion, the authors should clarify it in the manuscript. Without that, the current version leaves some doubt about the completeness of the study.

- We have now included a detailed explanation of focusing solely on evaluating AI-generated negative and neutral images (lines 140-151) from the theoretical, clinical and practical consideration in the introduction and discussed the potential limitation in the discussion section. We have also revised our study title to clarify that our evaluation of valence and arousal patterns is limited to the negative-neutral spectrum. As a first study, we prioritized evaluating negative versus neutral images due to the theoretical and clinical importance of negative affect processing. Including both negative and positive images in the same study would inevitably increase the total number of images participants needed to view and rate (for around 50% more images), potentially compromising the reliability of the ratings due to fatigue and habituation. After weighing the pros and cons of different designs, we decided to focus on the AI-generated negative image in this study and acknowledged in the limitations section that this methodological constraint limited the generalizability of our results (lines 569-574).

Reviewer #3:

1. Despite these strengths, the study has several serious limitations that must be addressed before it can be considered for publication. Most notably, the study excludes positive stimuli entirely, undermining the central claim that AI-generated images can replicate the full range of affective responses typically captured in standardized image sets. Without positive images, the U-shaped valence–arousal distribution is only partially validated, which significantly limits the generalizability of the findings. The authors must clearly justify this omission and discuss its implications. … While the authors briefly acknowledge the absence of positive images and the choice to include AI-distorted outputs, these decisions require more detailed justification and discussion. Specifically, the complete omission of positive stimuli significantly limits the ability to validate the full valence–arousal spectrum and undermines the claim that AI-generated images replicate the distributional patterns observed in standardized databases like IAPS. The rationale for this exclusion is not sufficiently developed.

- We thank the reviewer for highlighting this important issue. We agree that excluding positive images limits our ability to characterize the full valence–arousal distribution observed in standardized databases. In response, we have (i) added a more detailed rationale in the Introduction (lines 140-151) for focusing on AI-generated negative and neutral images, drawing on theoretical, clinical, and practical considerations; and (ii) explicitly discussed this limitation in the Discussion (lines 569-574). We have also revised the title of the manuscript to clarify that our evaluation of valence–arousal patterns is restricted to the negative–neutral spectrum.

As an initial study, we prioritized negative versus neutral affect given the theoretical and clinical salience of negative affect processing. Including positive images in the same protocol would have increased the number of stimuli by approximately 50%, raising risks of participant fatigue and habituation and, in turn, potentially reducing rating reliability. After weighing these trade-offs, we elected to focus on AI-generated negative (and neutral) images in this study and to acknowledge the resulting limits on generalizability in the Limitations section (lines 569-572). We view a balanced inclusion of positive stimuli as a key direction for subsequent work designed specifically to test the full valence–arousal space.

2. Second, no data were collected on whether participants recognized the images as AI-generated. This omission is problematic, as prior work suggests that perceived

artificiality can modulate emotional responses. Furthermore, the authors report that images with visual distortions or unrealistic elements were intentionally retained. While this decision may reflect practical constraints, it raises concerns about ecological validity and the applicability of findings to real-world emotional processing. Additionally, although the authors note that realism was not a primary goal, they do not assess whether participants were aware that the images were AI-generated or whether such awareness might have influenced their emotional ratings. Given growing evidence that perceived authenticity can modulate affective responses, this represents a potential confound that should be directly addressed. Both points—the exclusion of positive stimuli and the lack of participant awareness checks—require clearer theoretical and methodological justification.

- We did not formally obtain data on whether participants recognized the images as AI-generated. However, informally at the end of each study, we asked participants about their initial perception of the stimuli, but this data is not standardized enough to be included in our analysis. Whether recognizing the origin of the images will influence emotional responses is not clear cut. Some recent studies (Zhou & Kawabata, 2023; Bilucaglia et al., 2025) have indicated implicit effects of participants’ awareness, but this effect does not extend to explicit subjective ratings of valence and arousal. This discussion is now included in the manuscript in the limitations section (see lines 574-586)

The main purpose of the AI-generated images was for emotional elicitation in the context of typical experimental tasks in affective research. Considering that the presentation time of the images in these tasks is fairly short (e.g., few seconds), we did not try to modify the generated images to ensure that they were perceived as ‘real’. This has been explained in the method section (lines 231-238) and also in the discussion (lines 559-568)

3. The paper provides only a few examples of the generated images and their corresponding prompts, which raises concerns about transparency and reproducibility. Given that the study’s main contribution lies in the use of generative AI for flexible stimulus creation, it is essential to disclose a representative and sufficiently detailed sample of prompts, along with generated outputs. Without this information, it is difficult to assess the thematic accuracy, variability, or potential biases in the generation process. Most critically from a methodological standpoint, the authors do not address the inherent stochasticity of text-to-image generation models. Most generative AI systems sample from a probability distribution, meaning that identical prompts can yield different images across runs—especially on platforms like Adobe Firefly and DALL·E, which do not allow for control over random seed values or generation parameters. This severely compromises the reproducibility of the stimuli. The authors’ proposal that AI-generated images could replace standardized image databases is therefore premature unless procedures are put in place to ensure deterministic, shareable outputs. At a minimum, the authors should document the AI model configuration and any image metadata. If the same prompt was used in both Study 1 and Study 2, it would be valuable to examine and report whether the emotional ratings of the resulting images differed—highlighting the variability introduced by prompt reuse without seed control.

- While our findings showed that AI-generated images can be valid stimuli for affective research in our studies, we did not intend to replace standardized image databases with these AI-generated images. Instead, we aim to demonstrate the feasibility of generative AI as a powerful complementary tool that allows researchers to flexibly generate their own set of affective stimuli tailored to specific experiment requirements. We also show how to validate these AI-generated stimuli, either by the same group of participants after the main experiment (Study 1) or by a separate group of participants who were not exposed to the stimuli before the rating (Study 2). Although we never intend to suggest researchers replicate the same set of affective stimuli using the prompts we used, we’ve discussed the challenges of reproducibility in the manuscript (lines 553-558) and provided additional text prompts, generated outputs, and modified images from our dataset. These prompts represent those used across Adobe Firefly, Leonardo.ai, and Stable Diffusion platforms (see S1 Table, supplementary). While these prompts will not recreate the exact image, they still offer valu

---

## [Decision Letter · Decision Letter 1]

1 Dec 2025

Dear Dr. Wong,

Thank you for submitting your manuscript to PLOS ONE. After careful consideration, we feel that it has merit but does not fully meet PLOS ONE’s publication criteria as it currently stands. Therefore, we invite you to submit a revised version of the manuscript that addresses the points raised during the review process.

We look forward to receiving your revised manuscript.

Kind regards,

Sandra Carvalho, Ph.D.

Academic Editor

PLOS ONE

**Journal Requirements:**

Reviewers' comments:

Reviewer's Responses to Questions

**Comments to the Author**

Reviewer #1: All comments have been addressed

Reviewer #2: (No Response)

Reviewer #3: All comments have been addressed

2. Is the manuscript technically sound, and do the data support the conclusions?

Reviewer #1: Yes

Reviewer #2: Partly

Reviewer #3: Partly

3. Has the statistical analysis been performed appropriately and rigorously?

Reviewer #1: Yes

Reviewer #2: Yes

Reviewer #3: Yes

4. Have the authors made all data underlying the findings in their manuscript fully available?

Reviewer #1: Yes

Reviewer #2: Yes

Reviewer #3: No

5. Is the manuscript presented in an intelligible fashion and written in standard English?

Reviewer #1: Yes

Reviewer #2: Yes

Reviewer #3: Yes

**Reviewer #1:** The authors have responded to all of the reviewer's recommendations. I agree with the acceptance of the paper for publication.

**Reviewer #2:** Thank you for the revision incorporating my comments. The revised manuscript looks significantly more rigorous than the original one, which I appreciate. Of the two key limitations I mentioned, I think the issue of not testing positive images, which was also raised by another reviewer, has now been addressed as far as it can be taken at this point. That said, it seems that the first limitation—the potentially problematic inclusion of Study 1—has not been fully resolved.

In fact, the following response from the authors has crystallized that Study 1 has a significant methodological issue: "However, this design introduced a potential confound of habituation, as the ratings were collected upon the participants' second exposure to the images." Given this potential issue, and considering the nature of this research, retaining only Study 2 appears to be a more reasonable and scientifically sound choice than including a study that may be fundamentally compromised (i.e., Study 1). As a fellow researcher, I fully understand the desire to share as much work as possible, but I would like to encourage the authors to prioritize scientific rigor over the scale of their manuscript. In the end, I believe what we must prioritize as researchers is sharing the most reliable results/findings, thereby serving as a safe stepping stone for future research. if the authors would still like to stick to the dual-study approach, I suggest that they conduct an additional study to accompany Study 2 rather than retaining the current Study 1.

I sincerely hope my comments are helpful, and I wish the authors all the best in their continued work. Thank you.

**Reviewer #3:**  Point 1

Lines 140-151:

The rationale provided remains somewhat unconvincing. Specifically, the theoretical and clinical arguments (lines 140-145) explain why negative images are valuable, but do not justify excluding positive images. These are distinct questions.

The statement about sourcing negative images from databases being "more challenging" is confusing in this context. Since you are using AI to generate images rather than sourcing from existing databases, this point does not logically support your methodological choice.

Your strongest justification is participants' burden and task feasibility but it is buried at the end of this paragraph. The practical constraint should be the primary rationale presented.

I recommend restructuring this paragraph to lead with the practical constraints (task duration, fatigue, habituation), followed by the clinical relevance of negative stimuli. Either remove or significantly revise the database sourcing statement, as it undermines rather than strengthens your argument.

Lines 569-574:

The limitation is now acknowledged, which is good. However, the phrase "opposite end of the observed U-shaped distribution" is technically imprecise, as you have observed only the left portion of what may be a U-shaped distribution. Please revise to state that you cannot determine whether positive images would "complete" or "extend the observed pattern to form" the U-shaped distribution, and explicitly note that this limits claims about fully replicating distributional patterns from standardized databases.

As a general concern, please review your manuscript to ensure that claims about replicating the "full range" or "complete spectrum" of affective responses are modified to accurately reflect your study's scope (negative-to-neutral range only).

Point 2

Lines 574-586

You mention that you informally collected data on participant perceptions and have approximate numbers (11 in Study 1, 27 in Study 2). This represents a significant missed opportunity. If possible, include these informal observations in more detail, even if with appropriate caveats about the unsystematic collection method. A simple descriptive comparison of ratings from participants who noticed vs. didn't notice AI generation would strengthen the paper.

Regarding Zhou & Kawabata, 2023 and Bilucaglia et al., 2025, my understanding is that these studies actually present mixed findings: awareness affects some measures but not others. Their mixed evidence supports my original concern that awareness represents a potential confound that should have been measured and controlled. Please clarify which of your measures are not confounded by awareness with respect to the two papers' measures.

Point 3

Ok. However, I recommend that your negative and neutral AI stimuli be made publicly available on platforms such as OSF.io or similar repositories.

**Do you want your identity to be public for this peer review?** For information about this choice, including consent withdrawal, please see our Privacy Policy

Reviewer #1: No

Reviewer #2: No

Reviewer #3: No

---

## [Author Response · Author response to Decision Letter 2]

8 Jan 2026

We greatly appreciate the reviewer’s comments on the manuscript. Below, we have organized the comments that require revision and our point-by-point response to each comment.

Reviewer #1:

1. The authors have responded to all of the reviewer's recommendations. I agree with the acceptance of the paper for publication.

Thank you for your feedback and comments throughout the review process.

Reviewer #2:

1. …the potentially problematic inclusion of Study 1—has not been fully resolved. In fact, the following response from the authors has crystallized that Study 1 has a significant methodological issue: "However, this design introduced a potential confound of habituation, as the ratings were collected upon the participants' second exposure to the images." Given this potential issue, and considering the nature of this research, retaining only Study 2 appears to be a more reasonable and scientifically sound choice than including a study that may be fundamentally compromised (i.e., Study 1). As a fellow researcher, I fully understand the desire to share as much work as possible, but I would like to encourage the authors to prioritize scientific rigor over the scale of their manuscript. In the end, I believe what we must prioritize as researchers is sharing the most reliable results/findings, thereby serving as a safe stepping stone for future research. if the authors would still like to stick to the dual-study approach, I suggest that they conduct an additional study to accompany Study 2 rather than retaining the current Study 1.

Thank you for your thoughtful advice regarding the potential confound of habituation in Study 1. We have retained the Study 1 data because collecting ratings post-task is a standard practice in the field to verify that stimuli elicited the intended emotions during the specific experiment (a manipulation check). However, we fully agree with your concern that these ratings can be influenced by prior exposure.

Consequently, we have revised the Introduction and Discussion to explicitly address this limitation and emphasize the necessity of Study 2. We highlight that while Study 1 offers ecological validity regarding the task itself, Study 2 provides a cleaner, exposure-naïve benchmark (lines 147-157). As noted in the revised text (lines 508-510), the convergence of results across both studies demonstrates the robustness of the AI-generated images, regardless of exposure level. We believe reporting both datasets allows future researchers to assess the generalizability of these stimuli across different experimental contexts.

Reviewer #3:

1. The rationale provided remains somewhat unconvincing. Specifically, the theoretical and clinical arguments (lines 140-145) explain why negative images are valuable, but do not justify excluding positive images. These are distinct questions.

The statement about sourcing negative images from databases being "more challenging" is confusing in this context. Since you are using AI to generate images rather than sourcing from existing databases, this point does not logically support your methodological choice. Your strongest justification is participants' burden and task feasibility but it is buried at the end of this paragraph. The practical constraint should be the primary rationale presented. I recommend restructuring this paragraph to lead with the practical constraints (task duration, fatigue, habituation), followed by the clinical relevance of negative stimuli. Either remove or significantly revise the database sourcing statement, as it undermines rather than strengthens your argument.

We appreciate your thoughtful critique and agree that the rationale for focusing exclusively on negative images in Study 1 could be clarified. We have taken your recommendation to restructure the mentioned paragraph (lines 138-46) to emphasize the practical constraints as the driving rationale for excluding positive images, followed by the clinical relevance of negative stimuli. The sentence regarding the database sourcing challenges is now removed.

2. The limitation is now acknowledged, which is good. However, the phrase "opposite end of the observed U-shaped distribution" is technically imprecise, as you have observed only the left portion of what may be a U-shaped distribution. Please revise to state that you cannot determine whether positive images would "complete" or "extend the observed pattern to form" the U-shaped distribution, and explicitly note that this limits claims about fully replicating distributional patterns from standardized databases.

As a general concern, please review your manuscript to ensure that claims about replicating the "full range" or "complete spectrum" of affective responses are modified to accurately reflect your study's scope (negative-to-neutral range only).

We have revised the limitations section to emphasize that the interpretation of our findings is restricted to our scope of negative and neutral images only (lines 561-565). We have further reviewed the manuscript to ensure that readers are aware that the study scope is within the negative-to-neutral range and removed the use of the term ‘U-shaped’ in describing our study results to avoid confusion.

3. You mention that you informally collected data on participant perceptions and have approximate numbers (11 in Study 1, 27 in Study 2). This represents a significant missed opportunity. If possible, include these informal observations in more detail, even if with appropriate caveats about the unsystematic collection method. A simple descriptive comparison of ratings from participants who noticed vs. didn't notice AI generation would strengthen the paper.

We have now expanded our discussion of the informal observations to include the suggested descriptive comparison of ratings between participants who noticed the AI generation and those who did not. The discussion is now presented in the main text (lines 566-571) and the Supporting Information in S4 Table. Additionally, during a re-verification of our informal data logs, we identified a clerical error regarding the participant count in Study 2. The correct number of participants who noticed the AI generation is 19, not 27. We updated the manuscript accordingly. We sincerely apologize for this oversight and have subsequently re-verified all other numerical values to ensure accuracy.

4. Regarding Zhou & Kawabata, 2023 and Bilucaglia et al., 2025, my understanding is that these studies actually present mixed findings: awareness affects some measures but not others. Their mixed evidence supports my original concern that awareness represents a potential confound that should have been measured and controlled. Please clarify which of your measures are not confounded by awareness with respect to the two papers' measures.

Both Zhou & Kawabata (2023) and Bilucaglia et al. (2025) found that subjective ratings (i.e., valence/arousal) were not significantly affected by participants’ awareness of AI generation. This aligns with our measures in this study of self-reported valence and arousal ratings. Nevertheless, these studies have identified effects on physiological and behavioral measures (e.g. fixation times), suggesting that awareness of AI origins may exert implicit influence even when explicit ratings remain unchanged. Thus, we have clarified in the discussion that future research is required to understand how knowledge of AI origins affect different dimensions of emotional responses (lines 572-578).

5. Ok. However, I recommend that your negative and neutral AI stimuli be made publicly available on platforms such as OSF.io or similar repositories.

While we fully support open science practices, our preference is to make the stimuli available upon request, which is the typical norm for traditional affective stimuli databases (e.g., IAPS, Nencki). The controlled sharing of the stimuli we believe will allow better prevention of misuse and the maintenance of records of data access for accountability purposes.

---

## [Editor Report · Decision Letter 2]

22 Jan 2026

Beyond Traditional Stimuli: Validating AI-Generated Images on the Negative-Neutral Spectrum for Affective Research

PONE-D-25-25040R2

Dear Dr. Wong,

We’re pleased to inform you that your manuscript has been judged scientifically suitable for publication and will be formally accepted for publication once it meets all outstanding technical requirements.

Kind regards,

Sandra Carvalho, Ph.D.

Academic Editor

PLOS One

Additional Editor Comments (optional):

Thank you for your careful revisions. The manuscript now clearly demonstrates the methodological validity of AI-generated affective stimuli, supported by robust analyses across two independent studies. The work is transparent, appropriately cautious in its claims, and well aligned with the journal’s scope. I am pleased to inform you that the manuscript is accepted for publication.
---

## [Editor Report · Acceptance letter]

18 Aug 2025

PONE-D-25-25040R2

PLOS One

Dear Dr. Wong,

I'm pleased to inform you that your manuscript has been deemed suitable for publication in PLOS One. Congratulations! Your manuscript is now being handed over to our production team.

Kind regards,

on behalf of

Professor Sandra Carvalho

Academic Editor

PLOS One